# A Short Review of the Toxicity of Dentifrices—Zebrafish Model as a Useful Tool in Ecotoxicological Studies

**DOI:** 10.3390/ijms241814339

**Published:** 2023-09-20

**Authors:** Piotr Stachurski, Wojciech Świątkowski, Andrzej Ciszewski, Katarzyna Sarna-Boś, Agnieszka Michalak

**Affiliations:** 1Department of Paediatric Dentistry, Medical University of Lublin, 20-059 Lublin, Poland; 2Department of Oral Surgery, Medical University of Lublin, 20-059 Lublin, Poland; wojciechswiatkowski@umlub.pl; 3Department of Paediatric Orthopaedics and Rehabilitation, Medical University of Lublin, 20-093 Lublin, Poland; andrzej.ciszewski@umlub.pl; 4Department of Dental Prosthetics, Medical University of Lublin, 20-059 Lublin, Poland; katarzynasarnabos@umlub.pl; 5Independent Laboratory of Behavioral Studies, Medical University of Lublin, 20-059 Lublin, Poland; agnieszka.michalak@umlub.pl

**Keywords:** *Danio rerio*, fluoride, abrasive substances, detergents

## Abstract

This review aims to summarize the literature data regarding the effects of different toothpaste compounds in the zebrafish model. *Danio rerio* provides an insight into the mechanisms of the ecotoxicity of chemicals as well as an assessment of their fate in the environment to determine long-term environmental impact. The regular use of adequate toothpaste with safe active ingredients possessing anti-bacterial, anti-inflammatory, anti-oxidant, and regenerative properties is one of the most effective strategies for oral healthcare. In addition to water, a typical toothpaste consists of a variety of components, among which three are of predominant importance, i.e., abrasive substances, fluoride, and detergents. These ingredients provide healthy teeth, but their environmental impact on living organisms are often not well-known. Each of them can influence a higher level of organization: subcellular, cellular, tissue, organ, individual, and population. Therefore, it is very important that the properties of a chemical are detected before it is released into the environment to minimize damage. An important part of a chemical risk assessment is the estimation of the ecotoxicity of a compound. The zebrafish model has unique advantages in environmental ecotoxicity research and has been used to study vertebrate developmental biology. Among others, the advantages of this model include its external, visually accessible development, which allows for providing many experimental manipulations. The zebrafish has a significant genetic similarity with other vertebrates. Nevertheless, translating findings from zebrafish studies to human risk assessment requires careful consideration of these differences.

## 1. Introduction

The vast majority of people use toothpaste as a part of their daily routine. The regular use of adequate toothpaste with safe active ingredients possessing anti-bacterial, anti-inflammatory, anti-oxidant, and regenerative properties is one of the most effective strategies for the prevention and treatment of gingivitis, periodontal pathologies, and caries leading to teeth loss [1].

In addition to 20−42% water, a typical toothpaste consists of a variety of components, among which three are of predominant importance, i.e., abrasives, fluoride, and detergents [2]. These ingredients are carefully selected to serve functions such as cleaning teeth, preventing cavities, freshening breath, and maintaining overall oral health. However, some of these toothpaste ingredients have been found to have potential ecotoxicological effects when they find their way into aquatic ecosystems [3].

Abrasive agents, like calcium carbonate and silica, along with cleaning agents such as hydrated silica, are used in toothpaste to aid in removing dental plaque and stains from teeth. These ingredients can contribute to increased turbidity in water bodies and interfere with light penetration, potentially disrupting aquatic ecosystems by affecting photosynthesis and nutrient cycling [4]. Fluoride, often added as sodium fluoride or sodium monofluorophosphate, is a key ingredient for preventing tooth decay. In aquatic environments, excessive fluoride levels can lead to water contamination. High fluoride concentrations have been linked to adverse effects on aquatic organisms, including reduced growth, altered behavior, and disruption of reproductive processes. Ingredients like triclosan, an antibacterial compound, are included in some toothpaste formulations to combat oral bacteria. Triclosan and its transformation products can find their way into aquatic systems, where they may contribute to the development of antibiotic-resistant strains of bacteria and disrupt aquatic ecosystems’ microbial communities [5]. Surfactants like sodium lauryl sulfate (SLS) are added to toothpaste to create foaming and aid in the dispersion of toothpaste during brushing. In aquatic environments, surfactants can affect water surface tension, potentially leading to adverse effects on aquatic organisms like fish by impairing their natural behaviors [6]. Whitening agents like hydrogen peroxide are employed in toothpaste formulations to lighten tooth color. However, their introduction into aquatic environments can raise concerns about potential effects on aquatic organisms’ development and physiological processes due to their oxidative properties. Toothpaste often contains flavoring agents to improve taste. While these ingredients are generally considered safe for human use, their presence in wastewater can affect aquatic organisms’ feeding behavior and alter microbial communities in water bodies [7].

The diverse range of ingredients which are inter alia dentifrices ingredients, intended to enhance oral health, can have, in high amounts, unintended ecotoxicological consequences when they enter aquatic environments through wastewater. The reported effects range from disruptions in aquatic organism behavior and growth to alterations in microbial communities and ecosystem dynamics. As our understanding of these effects continues to grow, it becomes increasingly important to consider the potential environmental impacts of oral care products and adopt practices that mitigate their negative consequences on aquatic ecosystems.

The potential risks of exposure to excessive toothpaste ingredients extend beyond their immediate environmental impact. Ingestion or inhalation of toothpaste residues containing certain compounds, such as antibacterial agents or fluoride, can lead to unintended human exposure. While toothpaste is designed for oral use, accidental ingestion or misuse can occur, especially in young children. Ingesting excessive fluoride, for instance, can lead to dental fluorosis or systemic health issues. Additionally, the potential for antibiotic-resistant bacteria development due to the release of antibacterial agents from toothpaste residues into wastewater raises concerns about human health. Antibiotic-resistant strains could enter the environment through contaminated water, soil, or food, ultimately impacting human health by reducing the effectiveness of antibiotics for treating infections [8].

Toothpaste ingredients, once introduced into aquatic environments, can pose long-term risks. These substances have the potential to accumulate in sediments and biota over time, leading to chronic exposure for aquatic organisms and, subsequently, potential impacts on food chains. Accumulation of certain compounds, like triclosan, in aquatic organisms can result in biomagnification, where higher trophic levels may experience higher concentrations of these compounds. The effects of toothpaste ingredients can also extend to aquatic organisms’ reproductive and developmental processes, with potential consequences for population dynamics and long-term ecosystem stability [9].

Thus, it is very important that the toxic properties of a chemical are detected before it is released into the environment to minimize damage. An important part of a chemical risk assessment is the estimation of the ecotoxicity of a compound. The evaluation of the ecological safety of toothpaste is of great interest, also in terms of human health. For this purpose, the zebrafish (*Danio rerio*) model is successfully used worldwide.

The zebrafish (*Danio rerio*) model has emerged as a valuable tool in the field of ecotoxicological studies due to several distinct advantages it offers. Zebrafish are small aquatic organisms that are not only easy to maintain but also cost-effective for laboratory research. One notable advantage lies in their rapid developmental rate, facilitating the execution of studies within relatively short timeframes. Additionally, the transparency of zebrafish embryos and larvae provides researchers with a unique advantage—the ability to directly visualize internal organs, tissues, and even cellular processes, aiding in the observation of potential toxicological effects at various stages of development.

However, one of the most compelling attributes of the zebrafish model is its genetic similarity to humans. While zebrafish are obviously not humans, they share a surprising degree of genetic homology with humans, particularly in terms of basic biological processes and pathways. The zebrafish has a significant genetic similarity with other vertebrates, sharing approximately 70% with humans [10]. This genetic resemblance makes zebrafish an excellent surrogate for investigating the potential impacts of chemicals and pollutants on human health.

This model not only aids in understanding the effects of environmental stressors on aquatic organisms but also provides valuable data that can inform broader discussions about the potential impacts of such stressors on human health and ecosystems.

Furthermore, during the development of zebrafish, the teeth are attached to only the fifth gill arch and are arranged in transverse rows (front to back) and from the back to the abdomen, but there are no teeth in the mouth. The complete zebrafish dentition consists of three rows with five ventral teeth (central, V), four medial teeth (mediodorsal, MD), and two dorsal teeth (dorsal, D) on each side. The first tooth is visible approximately 2 days post-fertilization (dpf) in the 4 V position, followed immediately by the tooth germs at the 3 V and 5 V positions, and the last teeth develop at 12 and 16 dpf. First-generation teeth buds develop directly from the pharyngeal epithelium and by the orthodontin. Interestingly, the dentition develops very symmetrically on both sides of the pharnyx to 10 dpf [11].

While zebrafish serve as valuable models in toxicological studies, it is essential to recognize their limitations as human analogs. The effects observed in zebrafish may not always precisely predict the effects of these ingredients in humans. The mechanisms of absorption, distribution, metabolism, and excretion of substances can vary between species, potentially leading to different outcomes even when exposed to the same compounds. Also, the dosage and exposure levels that induce toxic effects in zebrafish might not directly correspond to those harmful for humans. Zebrafish are aquatic organisms with continuous exposure to the surrounding environment, whereas humans have distinct lifestyles and mechanisms for dealing with toxins. Zebrafish embryos and larvae are commonly used in toxicological studies due to their rapid development. However, this might not fully represent the long-term effects observed in humans over the course of years. As a result, translating findings from zebrafish studies to human risk assessment requires careful consideration of these differences [12].

This review aims to summarize the literature data regarding the ecotoxicological effects of different toothpaste compounds in the zebrafish model (Figure 1). *Danio rerio* provides an interesting insight into the mechanisms of toxicity of chemicals as well as an assessment of their fate in the environment to determine long-term environmental impact.

## 2. Search Strategy

The search strategy for the present review involved the Medline, Scopus, and Web of Science databases. Each database was searched using the terms “toothpaste agent” in combination with “zebrafish” and “toxicity”. The articles were verified as presented in Figure 2. Research articles published in English from 2012 to 2022 were eligible for inclusion with further selection.

## 3. Discussion

The diverse range of toothpaste ingredients, intended to enhance oral health, can have unintended ecotoxicological consequences when they excessively enter aquatic environments through wastewater. The reported effects range from disruptions in aquatic organism behavior and growth to alterations in microbial communities and ecosystem dynamics [13]. As our understanding of these effects continues to grow, it becomes increasingly important to consider the potential environmental impacts of oral care products and adopt practices that mitigate their negative consequences on aquatic ecosystems.

### 3.1. Fluoride

In the European Union, for example, the concentration of fluoride in toothpaste is typically limited to 0.145% (1450 ppm) for children older than 6 years and adult toothpaste, while toothpaste for children under 6 years of age usually has a lower concentration, around 0.1% (1000 ppm). This distinction is made to prevent potential overexposure to fluoride in younger children who might ingest toothpaste. In the United States, the Food and Drug Administration (FDA) has similar guidelines for fluoride concentration in toothpaste. Currently, the amount of toothpaste applied to the toothbrush is more important than the concentration of fluoride in the toothpaste. Modern recommendations are from 1000 ppm in children, and the amount of toothpaste is described as a little bit on the bristles, a grain of rice, or a pea [14]. 

In toothpaste, fluoride can be found in one of four forms: sodium fluoride (NaF), sodium monofluorophosphate (SMFP), stannous fluoride (SnF2), or amine fluoride (AmF). It can also be used as a combination of two active substances, e.g., NaF with SMFP and AmF with SnF2. The second combination is an essential one because AmF itself is unstable, which would severely limit its application. NaF reduces, more significantly than unstable AmF, the number of viable bacteria in the biofilm found on various oral surfaces after toothpaste application. Meanwhile, SnF2 shows significantly higher efficiency in eliminating live bacteria than NaF [15]. Moreover, toothpastes containing SnF2 are used to reduce dentine hypersensitivity to everyday irritating stimuli [16]. 

Fluoride is present in toothpaste in various concentrations. In the European Union market, the concentration of fluoride in products approved for trade in drugstores must not exceed 1500 ppm F^−^. Higher concentrations are only available in pharmacies, and in the UK are only accessible with prescription. With a view to balancing the benefits of fluoride toothpaste and the risk of fluorosis, the European Academy of Paediatric Dentistry (EAPD) recommends limiting fluoride to a grain of rice-sized portion of toothpaste in children under 6 years of age and a toothpaste containing 1000 ppm F^−^ up to 2 years of age [14]. 

Knowledge concerning the unquestionably desirable effects of fluoride on enamel development and its efficiency in aspects of caries prevention should be on a par with knowledge concerning fluoride toxicity. This problem has been commonly studied and described. Also, the biota in bodies of water is affected by deposited NaF, which is used as a pesticide and for industrial purposes. With the current use of *Danio rerio* in environmental toxicity studies, this model organism is a perfect tool to evaluate the adverse effects of fluoride in vertebrate developmental biology studies. It was shown that sodium fluoride exposure (18.599, 36.832 mg/L of fluoride for 30 and 60 days) significantly affects ovarian development, disrupts reproductive hormones, affects oogenesis, induces oxidative stress, and causes apoptosis through both external and internal pathways in the zebrafish ovary [17]. Moreover, fluoride can substantially inhibit the growth of zebrafish and specifically affect their reproductive system by impairing not only ovarian but also testicular structure, altering steroid hormone levels and expression of steroidogenic genes related to sex hormone synthesis in zebrafish [18]. In addition, fluoride significantly affected the secretion of thyroid hormones by altering the microstructure of the gland and changing the expression of genes that regulate their synthesis in male zebrafish [19]. Differential expression and activity of Nrf2 and other stress response genes were demonstrated in the liver of zebrafish after individual and combined exposure to the xenobiotics fluorine and arsenic [20]. Data clearly show that NaF exposure has significant effects on the induction of oxidative stress and alteration of gene expressions in the liver of female zebrafish [21]. It has been reported that reactive oxygen species levels are raised along with increased malondialdehyde levels and reduced glutathione levels in the brain of zebrafish [22]. Moreover, it was found that zebrafish exposed to 15 ppm NaF for 30 and 90 days post-fertilization showed liver histopathology including hyperplasia, cytoplasmic degeneration, and nuclear fragmentation [23]. 

### 3.2. Abrasive Substances

Among all groups of toothpaste ingredients, abrasives and polishing agents are the most important, both in terms of function and quantity [23]. They constitute from 25 to 50% of the toothpaste content. They perform the basic function in the cleaning process, mechanically removing dental plaque and discoloration of external origin. They are also responsible for the texture of the paste [24].

The most popular abrasives are calcium carbonate, calcium and magnesium hydroxides, silicon oxide, hydroxyapatite, or polymethacrylate. However, the effectiveness of these compounds depends not on their chemical composition, but rather on the shape and size of the grains contained in the preparation. A spherical shape is considered to be optimal. In turn, the grain size should not exceed 10 micrometers [25].

The properties of kinds of toothpaste approved for sale on the European market are laid down in the standard ISO 11609 [26] according to it, the optimal RDA (Relative Dentin Abrasion) value for toothpastes for everyday use is assumed to be in the range of 30–70. Nowadays, in the selection of toothpaste composition and production, a tendency to reduce abrasiveness without losing cleaning efficiency is noticeable. This may be mainly due to the increased use of high-performance abrasives such as hydrated silica [27]. Calcium carbonate nanoparticles (CaCO_3_-NPs) are promising materials for various industrial applications. It is necessary to understand their toxicological profile in biological systems as human and environmental exposure to CaCO_3_-NPs increases along with global manufacturing production. By analyzing the cytotoxicity of CaCO_3_-NPs on two cell lines (NIH 3T3 and MCF7), calcium carbonate nanoparticles were shown to be safe in vitro as they did not cause cell mortality or genotoxicity. In addition, zebrafish treated with CaCO_3-_NPs developed without any abnormality, confirming the safety and biocompatibility of this nanomaterial [28].

Abrasive substances in toothpastes approved for sale on the European market have certain size and shape standards. Currently, nanoparticles (NPs) are the most commonly used. One of them is a nanoparticle of calcium carbonate. Calcium carbonate nanoparticles were shown to be safe in vitro as they did not cause cell mortality or genotoxicity. Nanotechnology investigates materials at the nanoscale level (0.1–100 nm in diameter). There are many commercially available nanoproducts such as silver, silicon, titanium, zinc, and gold. They are used in a variety of applications and released to the environment. Titanium dioxide (TiO_2_) is one of the most commonly used NPs. The doses (TiO_2_) specified in the standards were safe for zebrafish, but significantly excessive doses showed autophagy and cell necrosis. Studies on TiO_2_ molecules have shown that a larger dimension than nano causes developmental abnormalities. Different nanoparticles such as the rare earth oxide, iron oxide, gold, silica, and carbon induce autophagy depending upon molecule size and dispersion. Currently, toothpastes have less abrasion, which does not affect the quality of cleaning.

Among various types of nanoparticles, silica nanoparticles (Si-NPs) have become popular as nanostructuring, drug delivery, and optical imaging agents. Si-NPs are highly stable and could bioaccumulate in the environment. Although toxicity studies of Si-NPs to human and mammalian cells have been reported, their effects on aquatic biota, especially fish, have not been significantly studied. Results from the studies on the effect of nanoparticles on zebrafish are generally consistent with similar studies on human and mouse cells that have been reported so far. Thus, fish cell lines could be valuable for screening emerging contaminants in aquatic environments including NPs through rapid high-throughput cytotoxicity bioassays [29].

There are many commercial nanoproducts such as silver, silicon, titanium, and gold. They have various applications and are commonly found in the environment. Titanium dioxide (TiO_2_) is one of the most commonly used NPs. TiO_2_-NPs are used in plant production and medicine, as well as the production of food, toothpaste, sunscreens, cosmetics, and in wastewater treatment. Studies on the effects of this compound at higher doses on zebrafish showed autophagy and necrosis in Sertoli cells, which consequently negatively affected the spermatogenic cells and testicular morphology of zebrafish [30].

Duan J. et al. [31] assessed that silica nanoparticles induced cytotoxicity as well as oxidative stress and apoptosis. Results showed that Si-NPs induced pericardia toxicity and caused bradycardia. Exposure to Si-NPs is a possible risk factor for the cardiovascular system, causing embryonic malformations, including pericardial edema, yolk sac edema, and tail and head malformation. The larval behavior testing showed that the total swimming distance was decreased in a dose-dependent manner. Si-NPs caused persistent effects on larval behavior [31]. This group of researchers continued to assess the effects of nanoparticles on the *Danio rerio* cardiovascular system [32].

In later studies, [33] explored the inflammation–coagulation response and thrombotic effects of Si-NPs in endothelial cells and zebrafish embryos. For in vitro study, swollen mitochondria and autophagosomes were observed in the ultrastructural analysis. The cytoskeleton organization was disrupted by Si-NPs in vascular endothelial cells. Their data demonstrated that Si-NPs could induce inflammation–coagulation response and thrombotic effects via the JAK1/TF signaling pathway.

Dumitrescu E. et al. [34] examined the effect of glycine functionalization on Si-NPs and investigated changes in viability and developmental defects in the organs of zebrafish embryos upon exposure. Si-NPs caused damage which was localized in the brain, heart, and liver of zebrafish embryos. Results illustrated that surface modification of non-lethal particles can create different toxicity outcomes in the organs of exposed zebrafish embryos.

The different doses of silicon dioxide nanoparticles (SiO_2_-NPs) were evaluated to understand the effects of sizes of NPs on their bioavailability and toxicity in zebrafish (*Danio rerio*) embryos (25, 50, and 100 mg/L of 15 or 30 nm SiO_2_-NPs for 5 days, respectively). The results showed that SiO_2_ could be readily up-taken by zebrafish, and the accumulation of SiO_2_ was significantly higher in 15 nm treatment groups compared to 30 nm SiO_2_-NP-treated groups. Furthermore, exposure to 15 nm SiO_2_-NPs at the concentration of 100 mg/L resulted in more significant changes in reactive oxygen species (ROS) levels, and perturbation of lipid peroxidative and antioxidant system than the same concentration of 30 nm SiO_2_-NPs, which indicates that small-sized nano-SiO_2_ evoked more severe oxidative stress in zebrafish larvae. Given the above, 15 nm SiO_2_-NPs were more likely to enter and accumulate in zebrafish larvae, thus causing more serious oxidative stress in vivo. These results may provide additional information on the fate and toxicities of different sizes of NPs [35].

Makkar H. et al. [36] assessed the biocompatibility of two commercially available dental materials, mineral trioxide aggregate (MTA) and Biodentine™. The biocompatibility analysis was performed in embryonic zebrafish with the help of standard toxicity assays measuring essential parameters such as survivability and hatching. The toxicity analysis showed a significant reduction in the hatching rate and survivability rates along with morphological malformations. ROS and apoptosis assay results revealed a greater biocompatibility of Biodentine™ as compared to that of MTA which was concentration-dependent. The study provides a new vision and standard in dental material sciences for assessing the biocompatibility of potential novel and commercially available dental materials. 

Pham DH. et al. [37] showed in their study that Si-NPs do not cause any developmental, cardio-, or hepatotoxicity, but they possess a potential risk to the neurobehavioral system. In this study, they investigated the toxicity of Si-NPs with diameters of 20, 50, and 80 nm using an in vivo zebrafish platform that analyzes multiple endpoints related to developmental, cardio-, hepato-, and neurotoxicity. Results showed that except for an acceleration in hatching time and alterations in the behavior of zebrafish embryos/larvae, silica NPs did not elicit any developmental defects, nor any cardio- or hepatotoxicity. The behavioral alterations were consistent for both embryonic photomotor and larval locomotor responses and were dependent on the concentration and the size of Si-NPs. As embryos and larvae exhibited a normal touch response and early hatching did not affect the larval locomotor response, the behavior changes observed are most likely the consequence of modified neuroactivity [38]. much earlier examined the behavior changes in *Danio rerio*. The results showed a concentration-dependent increase in behavioral neurotoxicity, mortality, and malformation among larvae treated with the SiO2 nanoparticles of 15 nm and 50 nm. A comparison of the 15 nm and 50 nm NPs by K-means clustering analysis demonstrated that the 15 nm NPs have a greater neurotoxic effect than the 50 nm NPs, with the 50 nm NPs exhibiting greater developmental toxicity on the zebrafish larvae than the 15 nm NPs.

### 3.3. Detergents

Sodium lauryl sulfate, often abbreviated as SLS, is present in most toothpaste as well as shampoos, scalp cosmetics, hair dyes, bleaches, shower gels, cleansers, make-up bases, liquid soaps, washing powders, oils, and bath salts. Although SLS can be extracted from coconuts, it is produced by chemical synthesis for use in industry [6].

SLS is the sodium salt of sodium dodecyl sulfuric acid and is classified in the cosmetic ingredient database as a denaturant, a surfactant detergent, an emulsifier, and a foaming agent [6]. The function of detergents in toothpaste is to lower the surface tension, and thus facilitate the removal of dental plaque. They also show a slight antibacterial effect and have an inhibiting effect on plaque build-up. In normal use, they have no clinically significant effect on hard tissues but may have an irritating effect on soft tissues. This in turn may lead to the exacerbation of ongoing periodontal diseases, as well as influence the formation and development of gingival recession and recurring ulceration [39].

Detergents may affect soft tissues in different ways. For instance, SLS, the anionic sodium dodecyl sulfate (SDS) or Betaine amphoteric surfactants, can cause necrosis of epithelial cells. In contrast, the non-ionic surfactant (Pluronic™) increases epithelial cell viability. At the same time, detergents may increase the activity of inflammatory factors such as TNF, IL-1β, and IL-8, which are known factors related to the persistence of periodontal inflammation. It should be noted, however, that studies on the effect of detergents on soft tissues were conducted in vitro, so they did not take into account the protective effect of saliva. Moreover, it should not be forgotten that the above-mentioned agents contained in toothpastes co-exist with other substances, which may limit their harmful effects. For example, triclosan, often found in toothpastes, has anti-inflammatory effects and may mitigate the irritating effects of SLS [24,39].

Currently, surfactants are widely distributed in the environment as organic pollutants, and their toxicity has attracted a lot of attention. Ref. [40] assessed the effect of SDS, cationic surfactant-dodecyldimethylbenzylammonium chloride (1227), and non-ionic surfactant-polyoxyethylene fatty alcohol (AEO) on the behavior of zebrafish larvae. Five behavioral parameters were recorded using a larval rest/wake assay, including rest total, number of rest bouts, rest bout length, total activity, and waking activity. The results revealed that 1227 and AEO at 1 μg/mL affected larval locomotor activity, and that SDS had no significant impact on larval behavior [40]. In addition, the toxicity assay of three surfactants on developing zebrafish embryos was also performed. All three surfactants induced concentration-dependent shorter body length compared to SDS and 1227. Furthermore, in situ hybridization showed dependent responses. Exposure to AEO resulted in smaller head size and smaller eye size, and the smaller head size could be associated with reduced EGR2 expression. Altered ntl expression showed that developmental retardation is due to inhibited cell migration and growth. These findings provide references for ecotoxicological evaluations of different types of surfactants and play a warning role in the use of surfactants [40].

### 3.4. Antibacterial Agents

In the oral cavity, like other areas of the gastrointestinal tract, there is a natural microflora, the presence of which gives the host several beneficial properties. However, in the absence of proper oral hygiene, dental plaque (biofilm) can build up beyond what is consistent with oral health. This shifts the balance of dominant bacteria away from those related to health. Such shifts may predispose the site to dental caries, gingivitis, or periodontal disease [41]. Possible strategies for maintaining the stability and beneficial properties of the natural oral microflora include improving oral hygiene, for example, by using products containing safe antimicrobial, anti-inflammatory, and antioxidant substances.

The presence of distinct microbes in the periodontal environment, e.g., Aggregatibacter actinomycetemcomitans (A.a.), Porphyromonas gingivalis (P.g.), Tannerella forsythensis (T.f.), Treponema denticola (T.d, Porphyromonas endodontalis (P.e.), Fusobacterium nucleatum (F.n.), and Prevotella intermedia (P.i.). Nonnenmacher C. et al. [42]; Lee HJ. et al. [43] has been associated with increased levels of host-produced pro-inflammatory cytokines, such as tumor necrosis factor α (TNF-α), interleukin 6 (IL-6), and interleukin 17A (IL17A) [44]. It has become common knowledge that infection-induced chronic inflammation is closely associated with an imbalance of reactive oxygen/nitrogen species and antioxidant defense, so-called oxidative stress [45,46].

Antimicrobial oral hygiene products include chlorhexidine, fluorides [47,48], xylitol [49,50], triclosan [48], and their combinations [51]. These compounds show antibacterial, anti-caries, and anti-inflammatory activity in vivo. However, their toxicity may be underestimated.

The most common chemical antiseptic in toothpaste is triclosan (5-chloro-2-(2,4-dichlorophenoxy)-phenol), which is still widely used not only in personal care products such as soaps, toothpaste, and deodorants but also in cleaning (detergents, disinfectants) and plastic products [52,53]. Therefore, triclosan can be a significant contaminant in the aquatic environment, even though it is rapidly degraded by photodegradation [54].

Once triclosan levels are detected in various human tissues such as adipose tissue, brain, and liver [55,56], studies on its long-term effects on human health have been undertaken [57]. Toxic effects of triclosan have been also extensively evaluated using zebrafish as an animal model [58,59,60,61]. Triclosan’s mechanisms of toxicity encompass a range of effects on zebrafish, including endocrine disruption, oxidative stress, microbiota imbalance, altered behavior, and developmental and reproductive effects. Understanding these mechanisms is crucial for assessing the potential harm of triclosan on zebrafish populations and broader aquatic ecosystems. It can act as an endocrine disruptor by binding to hormone receptors, particularly those associated with thyroid hormones. In zebrafish, disruptions in thyroid hormone signaling can lead to developmental abnormalities, hinder growth, and impact the timing of metamorphosis [62]. Triclosan can also induce oxidative stress within cells by generating reactive oxygen species (ROS), which are harmful molecules that can damage cell structures and DNA. In zebrafish, oxidative stress can result in cellular dysfunction, inflammation, and even cell death. This oxidative damage can affect various physiological processes, including organ function and tissue integrity [58]. Triclosan’s antimicrobial properties can extend beyond their intended use, affecting not only pathogenic bacteria but also beneficial microbial communities in aquatic environments. In zebrafish, exposure to triclosan can disturb the gut microbiota, which plays a vital role in digestion, nutrient absorption, and overall health. Imbalances in the microbiota can lead to various health issues, including impaired growth and weakened immunity [63]. Studies suggest that triclosan exposure can influence behavior and neurological function in aquatic organisms. In zebrafish, exposure to triclosan has been linked to alterations in swimming behavior, impaired neural development, and changes in neurotransmitter levels. These effects can impact zebrafish survival, predator–prey interactions, and overall ecosystem dynamics [64,65]. Disruption of hormone signaling can have significant consequences for reproductive and developmental processes in zebrafish. Exposure to triclosan has been associated with delayed hatching, altered embryonic development, and reduced fertility. These effects can impact zebrafish populations and have cascading effects on aquatic ecosystems [66]. There are more than 70 papers from the last 10 years concerning the evaluation of triclosan activity in the zebrafish model. It was revealed that triclosan disrupts the early stages of zebrafish by interfering with many developmental processes such as cartilage development, organogenesis, breeding, and changes in biomarker levels [67,68]. Furthermore, triclosan leads to craniofacial morphosis in zebrafish [69], and acute triclosan exposure induces subtle cardiotoxicity in developing fish [70]. Triclosan decreased zebrafish hatching rate and led to a series of malformations, such as cardiovascular malformation [68]. Additionally, otolith formation and eye and body pigmentation were disturbed along with growth restriction and pericardial edema [71]. Ninety-six-hour LC50 studies performed in zebrafish embryos and adults showed lethal concentrations of 0.42 and 0.34 mg/L, respectively [60]. Also, foraging efficiency was decreased [61].

Additionally, chronic triclosan exposure may cause biological genotoxicity, hepatotoxicity, immunotoxicity, neurotoxicity, and cardiotoxicity, as well as impairment of lipid metabolism [9,69]. Triclosan increased levels of cholinesterase, lactate dehydrogenase, and glutathione S-Transferase in zebrafish larvae but not adult fish. Furthermore, it was reported that triclosan impaired lipid metabolism homeostasis in zebrafish by enhancing the mRNA expression of lipid b-oxidation genes [60].

In behavioral studies, triclosan reduced swimming distance and increased freezing duration in 5 dpf zebrafish. Also, the anxiety level was augmented, which was suggested to result from decreased acetylcholinesterase (AChE) activity [64]. Decrease in acetylcholinesteras activity, together with the influence on myelin basic protein (MBP) and synapsin IIa (syn2a) genes after 4 days of treatment of triclosan, also resulted in motor neuron innervations in skeletal muscles and reduced touch-evoked escape response in zebrafish larvae [65]. Neurotoxic effects of triclosan may also result from an increase in oxidative stress processes, which has been demonstrated in the gill and ovary of zebrafish [72]. More detailed information about the toxicological profile of triclosan in zebrafish is presented in Table 1. 

Chlorhexidine has been commonly used in dental practice as an antiseptic agent since 1970. It is a highly bactericidal and bacteriostatic compound, and it has a stronger effect on Gram-positive bacteria than on Gram-negative bacteria *Enterobacteria*, *Porphyromonas gingivalis*, *Fusobacterium nucleatum*, as well as different species of *Actinomyces* and *Streptococcus*, including *Streptococcus mutans*, which is considered the main etiological agent of dental caries. Some Pseudomonas and Proteus strains, acid-fast bacilli, and bacterial spores are resistant to it. The antibacterial effect is related to the damage of the bacterial cell wall (increased permeability). Chlorhexidine binds to dental plaque and the oral mucosa and is gradually released, protecting against bacteria for a long time (8–12 h). It is also completely safe, although it can sometimes cause local hypersensitivity [83].

Although chlorhexidine is one of the most used biocides in the world, its toxicity to aquatic organisms is poorly understood. Only Jesus and co-workers evaluated its effects on zebrafish embryos [84]. The revealed toxicity of chlorhexidine on zebrafish after 96 h of incubation showed EC50 of 804.0 μg/L, whereas the 15 min EC50 is 1694.0 μg/L. Furthermore, early hatching as well as developmental abnormalities were observed. Moreover, among enzymatic biomarkers, cholinesterase activity was increased in chlorhexidine solutions at a range of concentrations of 80–900 μg/L. Only the highest concentration increased catalase without influence on glutathione-S-transferase and lactate dehydrogenase activities [84].

### 3.5. Whitening and Flavoring Agents

Teeth whitening is the most popular cosmetic dental procedure. It comes as no surprise, then, that whitening toothpaste is a popular choice for whitening teeth at home. This market need is fully understood and addressed by most toothpaste brands, which offer teeth-whitening product lines. This aside, we must notice that whitening ingredients are also commonly present in non-whitening products. Therefore, the whitening properties of toothpaste can be considered important and desired, whether playing a major or supportive role in our daily oral care routine or not. Teeth-whitening ingredients cover different abrasives and bleaching agents, also of herbal origin, which remove and prevent extrinsic stains [85]. Additionally, most ingredients used in toothpaste, especially fluoride and abrasives, are characterized by unpleasant tastes, which are covered by various flavoring agents. Flavoring agents cover non-sugar sweeteners (i.e., sorbitol, glycerol) or refreshing ingredients (i.e., menthol, eucalyptus), which give a cooling and refreshing effect [86]. Though toothpaste flavors are not used to induce any therapeutic effects, those of herbal origin possess additional bioactive properties which may be of help in keeping teeth and gums healthy. On the other hand, flavors are also responsible for most allergy-related adverse reactions to toothpaste [87]. Hence, flavoring agents, even if they do not play a significant part in maintaining oral hygiene, play a great role in consumer choice and acceptance. The following part discusses literature data on toxic effects on zebrafish of the two most studied compounds of whitening and flavoring agents, i.e., hydrogen peroxide and glycerol, respectively. Additionally, Table 2 summarizes the effects of other selected whitening and flavoring agents present in toothpaste on zebrafish survival, physiology, and behavior.

Glycerol, apart from covering bitter taste, also improves texture and prevents the loss of water and subsequent hardening of toothpaste [86]. It is one of the most common hydrophilic solvents, a humectant with cryoprotectant properties and a low level of toxicity, frequently used in pharmaceutical formulation and biomedical studies including sperm cryopreservation. However, glycerol (5–15%) has been shown to reduce by more than a half the motility of zebrafish sperm within 15 min of incubation, indicating a lack of suitability as a cryoprotectant in zebrafish [88]. Moreover, no zebrafish oocyte exposed for 30 min to 10% glycerol retained the ability to mature and subsequently be fertilized (0% survival). Altogether, it shows that approximately 10% glycerol possesses an inhibitory effect on both zebrafish male and female fertility potential, therefore significantly reducing the reproductive success of zebrafish [89]. Glycerol-induced toxicity has been also intensively studied in zebrafish larvae. Embryotoxic effects of glycerol are concentration-dependent with a strong correlation to embryo stage/larvae age and exposure time. Thus, 0.5% glycerol is a maximal concentration without an effect on embryo survival, when applied to four-cell-stage embryos (1 hpf) with subsequent 48 h exposure [90]. When incubation time is shortened to 24 h, the maximal tolerated concentration reaches 1.5%, and embryos treated within the first 24 hpf with 2.5% glycerol display multiple abnormalities including anterior–posterior axis truncation, u-shaped somites, and cardia bifida [91]. Moreover, embryos subjected to 5% glycerol between 36 and 48 h show a survival rate at the level of ~60% [90]. Accordingly, the older larvae are, the less vulnerable to higher concentrations of glycerol. Larvae at 5–7 dpf remain morphologically unaffected at the concentration of 2.5% after 24 h exposure. However, this concentration affected blood circulation and impaired motility, expressed as the lack of touch response [91]. Finally, adult zebrafish exposed for 10 days to a low concentration of glycerol (0.1%) showed decreased aggressive behavior and disturbed ability to interact with conspecifics [92]. Altogether, the data presented above demonstrate that a compound recognized as well-tolerated and non-toxic for humans may exhibit abundant toxic effects in fish; therefore, it should be considered as pollutive to the water environment.

Hydrogen peroxide (H_2_O_2_) is a common whitening agent which removes extrinsic stains, thereby lightening tooth color [85]. H_2_O_2_ concentrations in water systems may range from nanomolar to micromolar and originate from natural bioactivities of aquatic ecosystems, as well as pollution sources [93]. It is a potent oxidant, one of the ROS molecules, with a strong potential for toxicity in humans and animals. Unsurprisingly, H_2_O_2_ causes a significant lethality in zebrafish. Exposure to 1 mg/mL H_2_O_2_ led to embryo death within 32 h with yolk abnormalities and the tail deformed, while the remaining embryos had delayed development and tail deformation [94]. Moreover, 4 hpf zebrafish larvae exposed to 5 mM H_2_O_2_ for up to 96 hpf showed a high mortality rate, a significant increase in ROS production, and cardiotoxicity expressed as pericardial edema [95]. Finally, exposure to 1 mM H_2_O_2_ of zebrafish from 4 hpf to 96 hpf caused an increase in mortality rate (over 60%) and oxidative damage (loss of SOD and CAT activity), as well as a decrease in hatching rate and heart rate, accompanied by body malformations such as yolk sac edema and bent spine [96]. The reason that H_2_O_2_ has high toxicity is inevitably linked to its oxidative potential; this oxidant has been widely used in toxicological studies to induce reactive oxygen species (ROS) generation and cytotoxicity in the zebrafish model [97,98,99,100,101,102]. Interestingly, the impact of low concentrations of H_2_O_2_ on zebrafish behaviors has also been of scientific interest. Yoon H. et al. [93] studied changes in the behavior of zebrafish after short-term exposure to low concentrations of H_2_O_2_. It has been shown that the safe H_2_O_2_ concentration for both larval and adult zebrafish is 10 nM. Meanwhile, 100 nM H_2_O_2_ affected color preference in 5 dpf larval zebrafish, as well as decreased average velocity, average acceleration, active time, and total distance moved in larvae and adult fish [93].

**Table 2 ijms-24-14339-t002:** The effects of selected whitening and flavoring agents present in toothpaste on zebrafish survival, physiology, and behavior.

Agents	Effect	Comment	Reference
Whitening agents			
*Antarctic lichen*	General toxicity (mechanism: undetermined)	Exposure to extracts of *Amandinea* sp. and *Umbilicaria Antarctica* from 6 hpf to 120 hpf significantly reduced the survival rate in zebrafish larvae at the concentration of 200 μg/mL and higher.	[103]
*Cetraria islandica*	Inhibition of melanogenesis (mechanism: tyrosinase inhibition)	48 h exposure (from 8 to 56 hpf) to subtoxic concentrations of extracts from Cetraria islandica (44 µg/mL) reduced pigmentation in zebrafish.	[104]
*Letharia vulpine*	Inhibition of melanogenesis (mechanism: tyrosinase inhibition)	48 h exposure (from 8 to 56 hpf) to subtoxic concentrations of extracts from Letharia vulpine (30 µg/mL) reduced pigmentation in zebrafish.	[104]
*Lichen metabolites*	Hepatotoxicity (mechanism: undetermined)	Evernic acid (60.2 μM), vulpic acid (15.5 μM), and psoromic acid (3.6 μM) showed liver toxicity in a transgenic line of zebrafish with liver-specific expression (fabp10a:DsRed2) after 3 days of exposure from 6 dpf.	[105]
Flavouring agents			
Menthol	General toxicity (mechanism: undetermined)	72 h embryo exposure to menthol at 0.01 mg/mL and higher resulted in an increased mortality and malformation rate, and a decreased hatching rate.	[106]
	Nociception (mechanism: nociceptors stimulation)	Menthol (1.2 mM) induces acute immediate orofacial nociception behavior in adult zebrafish.	[107,108,109]
	Haemolysis (mechanism: related to prooxidant properties)	Menthol (180–200 μmol/L) induced brisk hemolysis in zebrafish G6PD deficiency model after 48 h exposure	[110]
Eucalyptus extracts	General toxicity (mechanism: related to Fe^3+^ presence)	Adult zebrafish exposed for 96 h to tannins (140 mg·L^−1^) from eucalyptus leaf leachate reached 100% cumulative mortality	[111]
	Haemolysis (mechanism: nociceptors stimulation)	Eucalyptus oil (1:5000 in fishwater) induces hemolytic phenotype in zebrafish G6PD deficiency model after 72 h exposure	[112]
	Hyperlocomotion (mechanism: related to irritant properties)	Biomass smoke condensates from *Eucalyptus globulus* (30 μg EOM/mL) elevates locomotor activity measured in the dark in 6 dpf zebrafish larvae after 60 min of exposure	[113]
Cinnamon extracts	General toxicity, Morphological abnormalities (mechanism: undetermined) Inhibition of angiogenesis (mechanism: linked with PKC-dependent phosphorylation of MAPK)	Exposure to cinnamon extracts from *Cinnamon zeylanicum* exhibited LC50 of 0.0508 mg/mL and caused gross morphological deformities (especially of the spine, tail, cartilage, heart, and jaw), abnormal heartbeat, and delayed hatching rate. Moreover, cinnamon extract concentration of 250 μg/mL inhibited angiogenesis after 16 h exposure from 6–8 hpf.	[114,115]
Cinnamaldehyde	Morphological abnormalities, Vascular malformations and cardiotoxicity, Decreased hatching rate (mechanism: undetermined)	Pure cinnamaldehyde induced toxicity in 3–4 dpf zebrafish (line Tg(Fli1:EGFP)) after exposure at around 6 hpf with a 50% effect concentration (EC50) of 34–41 µM.	[116]
	General toxicity, Neurotoxicity, Hypolocomotion (mechanism: neurotoxicity associated with increased oxidative stress)	The LD50 of Cinnamaldehyde was determined to be 8.362 mg/L in larval zebrafish exposed from 0 to 120 hpf.	[117]

Perspectives on mitigating the abovementioned toxic effects of toothpaste ingredients involve a multidimensional approach that encompasses both regulatory measures and innovative research. Zebrafish, alongside other relevant model organisms and advanced techniques, can play a crucial role in developing and evaluating these strategies. However, we are aware of the limitations of this model. While zebrafish offer valuable insights into potential toxicological effects, they are not perfect human analogs. The limitations in species differences, dose–response relationships, target tissues, and the complexity of human systems underscore the need for a holistic approach to toxicological research. Findings from zebrafish studies should be complemented by data from other model organisms and in vitro assays and be carefully interpreted when considering their relevance to human health and environmental risk assessment.

However, utilizing advanced techniques like high-throughput screening and in vitro assays can expedite the assessment of potential toxic effects of toothpaste ingredients. These methods can provide rapid insights into the effects of different compounds and formulations, potentially reducing the reliance on animal testing. Zebrafish and other models can be integrated into these strategies to validate the findings and assess the real-world implications.

## Figures and Tables

**Figure 1 ijms-24-14339-f001:**
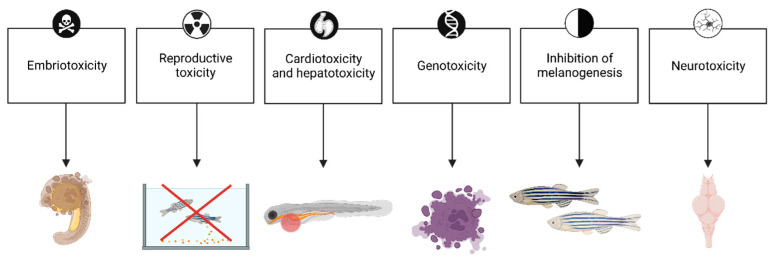
Revealed toxic effects of toothpaste ingredients in zebrafish. Created with BioRender.com (accessed on 15 April 2023).

**Figure 2 ijms-24-14339-f002:**
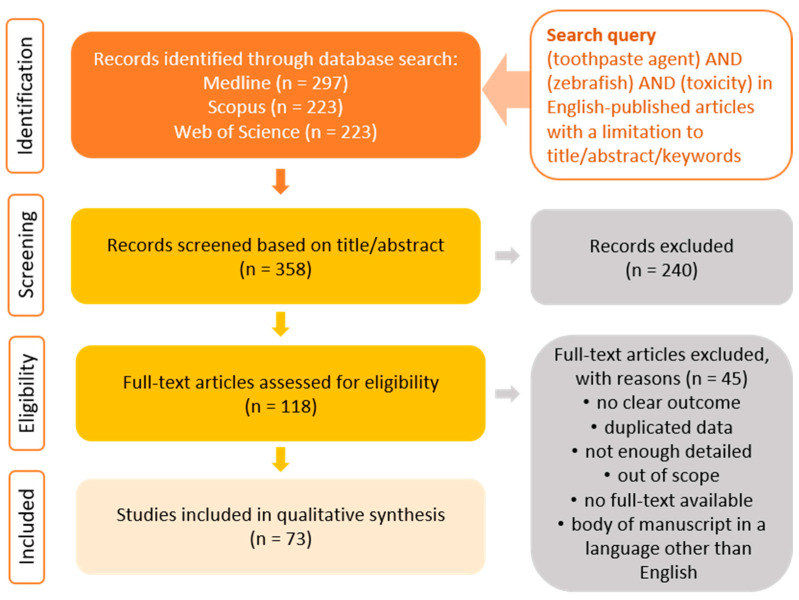
Prisma flow diagram illustrating the study selection process.

**Table 1 ijms-24-14339-t001:** The toxic effects of triclosan on zebrafish survival, physiology, and behavior.

Effect	Comment	Stage	Treatment (Concentration; Exposure Time)	Reference
Reproductive toxicity	Larval exposure to triclosan caused adverse effects in adults including delays in metamorphosis, as well as impairment of fecundity and fertility. Meanwhile, offspring were characterized by decreased survival and delayed maturation, without effect on reproductive capacity.	21–35 dpf larvae	40 μg/L; 15 days	[73]
General toxicity	Triclosan changes the expression of miRNAs involved in translation, transcription, and DNA-templated, protein transport, and motor neuron axon guidance.	2-month-old male zebrafish	68 μg/L for 42 days	[74]
Embryotoxicity Malformations	Recorded mortality and morphological changes in zebrafish embryos at 10 and 24 hpf.	2–24 hpf embryos	300 μg/L, 8 and 22 h	[75]
	Triclosan decreased the hatching rate in 72 hpf larvae, as well as caused a significant decrease in body length in 120 hpf larvae.	4–120 hpf larvae	300 μg/L, 4 to 120 hpf	[76]
	Triclosan induced craniofacial morphosis in zebrafish and decreased the body length, head size, and eye size in a concentration-dependent manner.	96 hpf larvae	0.2, 0.4, 0.6, and 0.8 mg/L; from 4 to 96 hpf	[69]
Cardiotoxicity	Incidence of pericardial edema, and impacts on heart structure and heart function.	8–120 hpf larvae	40, and 400 μg/L	[77]
Hepatotoxicity	TCS may be hepatotoxic in zebrafish; gene enrichment analysis further supported the role of the liver as a target organ for TCS toxicity.	6–48 hpf embryo	1–10 µM; from 8 to 120 hpf	[78]
Muscles	Trunk skeletal muscle abnormalities, presumably by the Ca^2+^ regulatory module between the dihydropyridine receptor and Ryanodine receptor 1.	96 hpf larvae	0.52, 1.04, and 1.73 μM; from 24 to 120 hpf	[79]
	Decreased acetylcholinesterase (AChE) activity in skeletal muscles, and the AChE gene was significantly downregulated only in the skeletal muscle, with observed downregulation of the myelin basic protein (MBP) gene.	Adult zebrafish (nine months old)	0.3 and 0.6 mg/L; for 48 h	[64]
Behavior	Triclosan reduced locomotion concomitant with increased freezing duration and induced anxiety-like behavior.	Adult zebrafish (nine months old)	0.3 and 0.6 mg/L; for 48 h	[65]
Changes in biomarkers	Significant dysregulations in the expression of the urea transporter (UT), glucose-6-phosphate dehydrogenase (G6PD), alanine transaminase (ALT), glutamate dehydrogenase (GDH), phosphoglucomutase (PGM), and fatty acid synthase (FASN), together with changes in alanine, urea, glucose, 6-phosphogluconalactone, and palmitic acid.	96 hpf larvae	30 μg/L and 300 μg/L; for 96 hpf	[80]
	Decrease in superoxide dismutase (SOD), catalase (CAT), and glutathione peroxidase (GPx) in the brain and liver of adult zebrafish. Also, the contents of the glutathione system (GSH and GSSH), as well as the activity of the glutathione reductase (GR), assayed in the liver, were reduced while the contents of malondialdehyde (MDA) were elevated in the liver.	Adult zebrafish (five months old)	50, 100, and 150 μg/L for 30 days	[81]
	Decrease in activity of glutathione-S-transferase (GST), P-glycoprotein efflux, and ethoxyresorufin-o-deethylase (EROD), and increase in oxidative stress parameters.	0–120 hpf larvae	0.1 μg/L and 1 μg/L; from 0 to 120 hpf; from 96 to 120 hpf	[82]

## Data Availability

Not applicable.

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
