# Peer review of "A Short Review of the Toxicity of Dentifrices—Zebrafish Model as a Useful Tool in Ecotoxicological Studies"

_ijms, 2023, doi:10.3390/ijms241814339_

Round 1
Reviewer 1 Report
Review report
Abstract
· The authors could have provided more information about the doses of toothpaste ingredients that were used in the studies that they reviewed. This information would be helpful for readers who are trying to assess the potential risks of toothpaste exposure to humans.
· The authors could have discussed the limitations of the zebrafish model in more detail. For example, zebrafish are not mammals, and their responses to toxicants may not be the same as those of humans.
· The authors could have provided more information about the potential for human exposure to toothpaste ingredients through other routes, such as ingestion or inhalation.
Introduction
· In the introduction, there is need to add a paragraph about the zebrafish model and its advantages for ecotoxicological studies. This paragraph could include the following information:
o Zebrafish are small, easy to care for, and relatively inexpensive to use.
o Zebrafish develop quickly, which allows for studies to be conducted over a short period of time.
o Zebrafish are transparent, which allows for researchers to visualize internal organs and tissues.
o Zebrafish are genetically similar to humans, which makes them a good model for studying the effects of chemicals on human health.
· In the literature review, need to organize the information into a more logical flow. For example, I would start by discussing the different types of toothpaste ingredients, then move on to the different ecotoxicological effects that have been reported for each ingredient. This would make it easier for readers to follow the flow of information and to identify the key findings of the article.
· In the discussion section, need to add a bit more discussion about the implications of the findings for human health and the environment. For example, I would discuss the potential risks of exposure to toothpaste ingredients through ingestion or inhalation. I would also discuss the potential for toothpaste ingredients to accumulate in the environment and to have long-term effects on aquatic ecosystems.
Discussion
· In the section on fluoride, the text could mention that the concentration of fluoride in toothpaste is regulated in the European Union and the United States. This information would be helpful for readers to understand the potential risks of exposure to fluoride.
· In the section on abrasives, the text could mention that the size and shape of abrasive particles can affect their toxicity. This information would be helpful for readers to understand how abrasive particles can harm zebrafish.
· In the section on nanoparticles, the text could mention that the toxicity of nanoparticles can depend on their size, shape, and surface charge. This information would be helpful for readers to understand how nanoparticles can harm zebrafish.
· The section on triclosan could be improved by providing more information about the doses of triclosan that were used in the studies, as well as the duration of exposure. This information would be helpful for readers to understand the potential risks of exposure to triclosan.
· The section on triclosan could also be improved by providing more information about the mechanisms of toxicity of triclosan. This information would be helpful for readers to understand how triclosan can harm zebrafish.
· The section on glycerol could be improved by providing more information about the doses of glycerol that were used in the studies, as well as the duration of exposure. This information would be helpful for readers to understand the potential risks of exposure to glycerol.
· The section on hydrogen peroxide could also be improved by providing more information about the doses of hydrogen peroxide that were used in the studies, as well as the duration of exposure. This information would be helpful for readers to understand the potential risks of exposure to hydrogen peroxide.
· The section on hydrogen peroxide could also be improved by providing more information about the mechanisms of toxicity of hydrogen peroxide. This information would be helpful for readers to understand how hydrogen peroxide can harm zebrafish.
· For the table 2, The table could be improved by including more information about the doses of the whitening and flavoring agents that were used in the studies, as well as the duration of exposure. This information would be helpful for readers to understand the potential risks of exposure to these agents.
· The table could also be improved by including more information about the mechanisms of toxicity of the whitening and flavoring agents. This information would be helpful for readers to understand how these agents can harm zebrafish.
The article needs a through revision in terms of language , gramatical mistakes and phrases
Author Response
Dear Reviewer,
We would like to express our sincere gratitude for your valuable feedback and insightful comments on our work. Your thoughtful input has been instrumental in enhancing the quality and clarity of our publication. Your expertise and dedication to the peer-review process are greatly appreciated.
Abstract
- The authors could have provided more information about the doses of toothpaste ingredients that were used in the studies that they reviewed. This information would be helpful for readers who are trying to assess the potential risks of toothpaste exposure to humans.
- The authors could have discussed the limitations of the zebrafish model in more detail. For example, zebrafish are not mammals, and their responses to toxicants may not be the same as those of humans.
The authors could have provided more information about the potential for human exposure to toothpaste ingredients through other routes, such as ingestion or inhalation.
Response: Thank you for the comment, the doses were added in the Discussion section because we should follow the editorial requirements regarding the number of characters in the Abstract Section.
Introduction
- In the introduction, there is need to add a paragraph about the zebrafish model and its advantages for ecotoxicological studies. This paragraph could include the following information:
o Zebrafish are small, easy to care for, and relatively inexpensive to use.
o Zebrafish develop quickly, which allows for studies to be conducted over a short period of time.
o Zebrafish are transparent, which allows for researchers to visualize internal organs and tissues.
- Zebrafish are genetically similar to humans, which makes them a good model for studying the effects of chemicals on human health.
Response: The following paragraphs have been added.
The zebrafish (Danio rerio) model has emerged as a valuable tool in the field of ecotoxicological studies due to several distinct advantages it offers. Zebrafish are small aquatic organisms that are not only easy to maintain but also cost-effective for laboratory research. One notable advantage lies in their rapid developmental rate, facilitating the execution of studies within relatively short timeframes. Additionally, the transparency of zebrafish embryos and larvae provides researchers with a unique advantage - the ability to directly visualize internal organs, tissues, and even cellular processes, aiding in the observation of potential toxicological effects at various stages of development.
However, one of the most compelling attributes of the zebrafish model is its genetic similarity to humans. While zebrafish are obviously not humans, they share a surprising degree of genetic homology with human, particularly in terms of basic biological processes and pathways. Zebrafish has a significant genetic similarity with other vertebrates sharing approximately 70 % with humans [Sarmah S. and Marrs JA., 2016].This genetic resemblance makes zebrafish an excellent surrogate for investigating the potential impacts of chemicals and pollutants on human health.
This model not only aids in understanding the effects of environmental stressors on aquatic organisms but also provides valuable data that can inform broader discussions about the potential impacts of such stressors on human health and ecosystems.
- In the literature review, need to organize the information into a more logical flow. For example, I would start by discussing the different types of toothpaste ingredients, then move on to the different ecotoxicological effects that have been reported for each ingredient. This would make it easier for readers to follow the flow of information and to identify the key findings of the article.
Response: Thank you for the comment. The section has been reorganized as follows:
Toothpaste is a widely used oral care product that contains a variety of ingredients aimed at promoting oral hygiene and health. These ingredients are carefully selected to serve functions such as cleaning teeth, preventing cavities, freshening breath, and maintaining overall oral health. However, some of these toothpaste ingredients have been found to have potential ecotoxicological effects when they find their way into aquatic ecosystems. Abrasive agents, like calcium carbonate and silica, along with cleaning agents such as hydrated silica, are used in toothpaste to aid in removing dental plaque and stains from teeth. These ingredients can contribute to increased turbidity in water bodies and interfere with light penetration, potentially disrupting aquatic ecosystems by affecting photosynthesis and nutrient cycling. Fluoride, often added as sodium fluoride or sodium monofluorophosphate, is a key ingredient for preventing tooth decay. In aquatic environments, excessive fluoride levels from toothpaste runoff can lead to water contamination. High fluoride concentrations have been linked to adverse effects on aquatic organisms, including reduced growth, altered behavior, and disruption of reproductive processes. Ingredients like triclosan, an antibacterial compound, are included in some toothpaste formulations to combat oral bacteria. Triclosan and its transformation products can find their way into aquatic systems, where they may contribute to the development of antibiotic-resistant strains of bacteria and disrupt aquatic ecosystems' microbial communities. Surfactants like sodium lauryl sulfate are added to toothpaste to create foaming and aid in the dispersion of toothpaste during brushing. In aquatic environments, surfactants can affect water surface tension, potentially leading to adverse effects on aquatic organisms like fish by impairing their natural behaviors. Whitening Agents like hydrogen peroxide are employed in toothpaste formulations to lighten tooth color. However, their introduction into aquatic environments can raise concerns about potential effects on aquatic organisms' development and physiological processes due to their oxidative properties. Toothpaste often contains flavoring agents and sweeteners to improve taste. While these ingredients are generally considered safe for human use, their presence in wastewater can affect aquatic organisms' feeding behavior and alter microbial communities in water bodies. Surfactants like SLS are added to toothpaste to create foaming and aid in the distribution of toothpaste during brushing. However, in aquatic environments, these surfactants can lower water surface tension, potentially impacting aquatic organisms' behaviors, including fish by impairing their natural swimming patterns”.
It was also addend at the beggining of discussion „The diverse range of toothpaste ingredients, intended to enhance oral health, can have unintended ecotoxicological consequences when they enter aquatic environments through wastewater. The reported effects range from disruptions in aquatic organism behavior and growth to alterations in microbial communities and ecosystem dynamics. As our understanding of these effects continues to grow, it becomes increasingly important to consider the potential environmental impacts of oral care products and adopt practices that mitigate their negative consequences on aquatic ecosystems”.
- In the discussion section, need to add a bit more discussion about the implications of the findings for human health and the environment. For example, I would discuss the potential risks of exposure to toothpaste ingredients through ingestion or inhalation. I would also discuss the potential for toothpaste ingredients to accumulate in the environment and to have long-term effects on aquatic ecosystems.
Response: The following paragraghs have been added.
The potential risks of exposure to toothpaste ingredients extend beyond their immediate environmental impact. Ingestion or inhalation of toothpaste residues containing certain compounds, such as antibacterial agents or fluoride, can lead to unintended human exposure. While toothpaste is designed for oral use, accidental ingestion or misuse can occur, especially in young children. Ingesting excessive fluoride, for instance, can lead to dental fluorosis or systemic health issues. Additionally, the potential for antibiotic-resistant bacteria development due to the release of antibacterial agents from toothpaste residues into wastewater raises concerns about human health. Antibiotic-resistant strains could enter the environment through contaminated water, soil, or food, ultimately impacting human health by reducing the effectiveness of antibiotics for treating infections.
Toothpaste ingredients, once introduced into aquatic environments, can pose long-term risks. These substances have the potential to accumulate in sediments and biota over time, leading to chronic exposure for aquatic organisms and, subsequently, potential impacts on food chains. Accumulation of certain compounds, like triclosan, in aquatic organisms can result in biomagnification, where higher trophic levels may experience higher concentrations of these compounds.The effects of toothpaste ingredients can also extend to aquatic organisms' reproductive and developmental processes, with potential consequences for population dynamics and long-term ecosystem stability.
Discussion
- In the section on fluoride, the text could mention that the concentration of fluoride in toothpaste is regulated in the European Union and the United States. This information would be helpful for readers to understand the potential risks of exposure to fluoride.
Response: This information was included in the manuscript.
In the European Union, for example, the concentration of fluoride in toothpaste is typically limited to 0.145% (1450 ppm) for children older than 6 years and adult toothpaste, while toothpaste for children under 6 years of age usu-ally has a lower concentration, around 0.1% (1000 ppm). This distinction is made to prevent po-tential overexposure to fluoride in younger children who might ingest toothpaste. In the United States, the Food and Drug Administration (FDA) has similar guidelines for fluoride concentration in toothpaste. Currently, the amount of toothpaste applied to the toothbrush is more important than the concentration of fluoride in the toothpaste Modern recommendations are from 1000ppm in children and the amount of toothpaste like a little bit on the bristles, a grain of rice, a grain of peas [Toumba et al., 2019].
- In the section on abrasives, the text could mention that the size and shape of abrasive particles can affect their toxicity. This information would be helpful for readers to understand how abrasive particles can harm zebrafish.
- In the section on nanoparticles, the text could mention that the toxicity of nanoparticles can depend on their size, shape, and surface charge. This information would be helpful for readers to understand how nanoparticles can harm zebrafish.
Response: Thank you for the comments the informations were added.
Abrasive substances in toothpastes approved for sale on the European market have certain size and shape standards. Currently, nanoparticles are the most commonly used. One of them is a nanoparticle of calcium carbonate.
Calcium carbonate nanoparticles were shown to be safe in vitro as they did not cause cell mortality or genotoxicity.
Nanotechnology investigates materials at nanoscale level (0.1-100nm in diameter).
There are many commercially nanoproducts such as silver, silicon, titanium, zinc, and gold. They are used in a variety of applications and released to the environment. Titanium dioxide (TiO2) is one of the most commonly used nanoparticles (NP). The doses (TiO2) specified in the standards were safe for zebra fish, but significant exceeding the doses showed autophagy and cell necrosis. Studies on TiO2 molecules have shown that a larger dimension than nano causes developmental abnormalities
Different nanoparticles such as the rare earth oxide, iron oxide, gold, silica and carbon induce autophagy depending upon molecule size and dispersion.
Currently, toothpastes have less abrasion, which does not affect the quality of cleaning.
The section on triclosan could be improved by providing more information about the doses of triclosan that were used in the studies, as well as the duration of exposure. This information would be helpful for readers to understand the potential risks of exposure to triclosan.
Response: Thank you for the comments, the doses and time of exposure of triclosan was added.
The section on triclosan could also be improved by providing more information about the mechanisms of toxicity of triclosan. This information would be helpful for readers to understand how triclosan can harm zebrafish.
Response: Thank you for the comments, the informations were added.
Triclosan's mechanisms of toxicity encompass a range of effects on zebrafish, including endocrine disruption, oxidative stress, microbiota imbalance, altered behavior, and developmental and reproductive effects. Understanding these mechanisms is crucial for assessing the potential harm of triclosan on zebrafish populations and broader aquatic ecosystems. It can act as an endocrine disruptor by binding to hormone receptors, particularly those associated with thyroid hormones. In zebrafish, disruptions in thyroid hormone signaling can lead to developmental abnormalities, hinder growth, and impact the timing of metamorphosis. Triclosan can also induce oxidative stress within cells by generating reactive oxygen species (ROS), which are harmful molecules that can damage cell structures and DNA. In zebrafish, oxidative stress can result in cellular dysfunction, inflammation, and even cell death. This oxidative damage can affect various physiological processes, including organ function and tissue integrity Triclosan's antimicrobial properties can extend beyond their intended use, affecting not only pathogenic bacteria but also beneficial microbial communities in aquatic environments. In zebrafish, exposure to triclosan can disturb the gut microbiota, which plays a vital role in digestion, nutrient absorption, and overall health. Imbalances in the microbiota can lead to various health issues, including impaired growth and weakened immunity .Studies suggest that triclosan exposure can influence behavior and neurological function in aquatic organisms. In zebrafish, exposure to triclosan has been linked to alterations in swimming behavior, impaired neural development, and changes in neurotransmitter levels. These effects can impact zebrafish survival, predator-prey interactions, and overall ecosystem Dynamics. Disruption of hormone signaling can have significant consequences for reproductive and developmental processes in zebrafish. Exposure to triclosan has been associated with delayed hatching, altered embryonic development, and reduced fertility. These effects can impact zebrafish populations and have cascading effects on aquatic ecosystems.
- The section on glycerol could be improved by providing more information about the doses of glycerol that were used in the studies, as well as the duration of exposure. This information would be helpful for readers to understand the potential risks of exposure to glycerol.
Response: All missing information has been added and highlighted in red.
- The section on hydrogen peroxide could also be improved by providing more information about the doses of hydrogen peroxide that were used in the studies, as well as the duration of exposure. This information would be helpful for readers to understand the potential risks of exposure to hydrogen peroxide.
Response: All missing information has been added and highlighted in red.
- The section on hydrogen peroxide could also be improved by providing more information about the mechanisms of toxicity of hydrogen peroxide. This information would be helpful for readers to understand how hydrogen peroxide can harm zebrafish.
Response: All missing information has been added and highlighted in red.
- For the table 2, The table could be improved by including more information about the doses of the whitening and flavoring agents that were used in the studies, as well as the duration of exposure. This information would be helpful for readers to understand the potential risks of exposure to these agents.
Response: All missing information has been added and highlighted in red.
- The table could also be improved by including more information about the mechanisms of toxicity of the whitening and flavoring agents. This information would be helpful for readers to understand how these agents can harm zebrafish.
Response: All missing information has been added and highlighted in red.

Reviewer 2 Report
In this article, Stachurski and colleagues thoroughly examine the potentially toxic effects of various toothpaste ingredients on zebrafish. They delve into the influence of these substances on zebrafish survival, physiology, and behavior, offering a valuable perspective on the potential health and environmental risks associated with these widely used substances. Despite the article's strengths, several areas of concern need to be addressed:
- The article could be better organized for enhanced readability. Subheadings to distinguish the discussion of each ingredient would make the article flow well for the readers. Furthermore, a summary or conclusion section would be beneficial in consolidating the key findings and implications of the review.
- The article could benefit from a comparative analysis of the relative toxicities of the different toothpaste ingredients. This would give readers a clearer understanding of which substances pose the greatest risks.
- The authors should include a discussion of their perspectives on the potential strategies for mitigating the toxic effects of these ingredients and how zebrafish and/or other models could be utilized for this.
- The authors should conclude with the limitations of their study.
- While zebrafish are commonly used in toxicological studies, they are not perfect human analogs. As such, the effects observed in zebrafish may not always accurately predict the effects of these ingredients in humans.
- The article should include more recent studies to ensure that the review is current with the latest research in this field. The majority of the references in the current version are from older studies.
There are several grammatical errors including spelling, tenses, spacing, and punctuation throughout the article. A professional proofreading is highly recommended.
Author Response
Dear Reviewer,
We would like to express our sincere gratitude for your valuable feedback and insightful comments on our work. Your thoughtful input has been instrumental in enhancing the quality and clarity of our publication. Your expertise and dedication to the peer-review process are greatly appreciated.
In this article, Stachurski and colleagues thoroughly examine the potentially toxic effects of various toothpaste ingredients on zebrafish. They delve into the influence of these substances on zebrafish survival, physiology, and behavior, offering a valuable perspective on the potential health and environmental risks associated with these widely used substances. Despite the article's strengths, several areas of concern need to be addressed:
- The article could be better organized for enhanced readability. would make the article flow well for the readers. Furthermore, a summary or conclusion section would be beneficial in consolidating the key findings and implications of the review.
Response: Thank you for the comment Subheadings distinguishing in the discussion the each ingredient are added. The section has been reorganized as follows:
„Perspectives on mitigating of abovementioned the toxic effects of toothpaste ingredients involve a multidimensional approach that encompasses both regulatory measures and innovative research. Zebrafish, alongside other relevant model organisms and advanced techniques, can play a crucial role in developing and evaluating these strategies. However, we are aware of the limitations of this model. While zebrafish offer valuable insights into potential toxicological effects, they are not perfect human analogs. The limitations in species differences, dose-response relationships, target tissues, and the complexity of human systems underscore the need for a holistic approach to toxicological research. Findings from zebrafish studies should be complemented by data from other model organisms and in vitro assays and be carefully interpreted when considering their relevance to human health and environmental risk assessment.
However, utilizing advanced techniques like high-throughput screening and in vitro assays can expedite the assessment of potential toxic effects of toothpaste ingredients. These methods can provide rapid insights into the effects of different compounds and formulations, potentially reducing the reliance on animal testing. Zebrafish and other models can be integrated into these strategies to validate the findings and assess the real-world implications.”
- The article could benefit from a comparative analysis of the relative toxicities of the different toothpaste ingredients. This would give readers a clearer understanding of which substances pose the greatest risks.
The paragfaph was addend „However, some of these toothpaste ingredients have been found to have potential ecotoxicological effects when they find their way into aquatic ecosystems. Abrasive agents, like calcium carbonate and silica, along with cleaning agents such as hydrated silica, are used in toothpaste to aid in removing dental plaque and stains from teeth. These ingredients can contribute to increased turbidity in water bodies and interfere with light penetration, potentially disrupting aquatic ecosystems by affecting photosynthesis and nutrient cycling [Mitsui T., 1997]. Fluoride, often added as sodium fluoride or sodium monofluorophosphate, is a key ingredient for preventing tooth decay. In aquatic environments, excessive fluoride levels from toothpaste runoff can lead to water contamination. High fluoride concentrations have been linked to adverse effects on aquatic organisms, including reduced growth, altered behavior, and disruption of reproductive processes. Ingredients like triclosan, an antibacterial compound, are included in some toothpaste formulations to combat oral bacteria. Triclosan and its transformation products can find their way into aquatic systems, where they may contribute to the development of antibiotic-resistant strains of bacteria and disrupt aquatic ecosystems' microbial communities [Menendez A. et al, 2005].. Surfactants like sodium lauryl sulfate ( SLS) are added to toothpaste to create foaming and aid in the dispersion of toothpaste during brushing. In aquatic environments, surfactants can affect water surface tension, potentially leading to adverse effects on aquatic organisms like fish by impairing their natural behaviors. Whitening Agents like hydrogen peroxide are employed in toothpaste formulations to lighten tooth color. However, their introduction into aquatic environments can raise concerns about potential effects on aquatic organisms' development and physiological processes due to their oxidative properties. Toothpaste often contains flavoring agents to improve taste. While these ingredients are generally considered safe for human use, their presence in wastewater can affect aquatic organisms' feeding behavior and alter microbial communities in water bodies [Khamverdi Z. et al, 2010]. Surfactants like SLS are added to toothpaste to create foaming and aid in the distribution of toothpaste during brushing. However, in aquatic environments, these surfactants can lower water surface tension, potentially impacting aquatic organisms' behaviors, including fish by impairing their natural swimming patterns [Healy CM. et al, 2000].
- The authors should include a discussion of their perspectives on the potential strategies for mitigating the toxic effects of these ingredients and how zebrafish and/or other models could be utilized for this.
Response: Thank you for the valuable comment. We added the informations at the Discussion section „Perspectives on mitigating of abovementioned the toxic effects of toothpaste ingredients involve a multidimensional approach that encompasses both regulatory measures and innovative research. Zebrafish, alongside other relevant model organisms and advanced techniques, can play a crucial role in developing and evaluating these strategies. (…..)However, utilizing advanced techniques like high-throughput screening and in vitro assays can expedite the assessment of potential toxic effects of toothpaste ingredients. These methods can provide rapid insights into the effects of different compounds and formulations, potentially reducing the reliance on animal testing. Zebrafish and other models can be integrated into these strategies to validate the findings and assess the real-world implications.”
- The authors should conclude with the limitations of their study.
Response: Thank you for the valuable comment. We added the limitations at the Discussion section „While zebrafish offer valuable insights into potential toxicological effects, they are not perfect human analogs. The limitations in species differences, dose-response relationships, target tissues, and the complexity of human systems underscore the need for a holistic approach to toxicological research. Findings from zebrafish studies should be complemented by data from other model organisms and in vitro assays and be carefully interpreted when considering their relevance to human health and environmental risk assessment.”
- While zebrafish are commonly used in toxicological studies, they are not perfect human analogs. As such, the effects observed in zebrafish may not always accurately predict the effects of these ingredients in humans.
Response: Thank you for the valuable comment. We addend the informations at the Introduction section „While zebrafish serve as valuable models in toxicological studies, it's essential to recognize their limitations as human analogs. The effects observed in zebrafish may not always precisely predict the effects of these ingredients in humans. The mechanisms of absorption, distribution, metabolism, and excretion of substances can vary between species, potentially leading to different outcomes even when exposed to the same compounds. Also the dosage and exposure levels that induce toxic effects in zebrafish might not directly correspond to those harmful for humans. Zebrafish are aquatic organisms with continuous exposure to the surrounding environment, whereas humans have distinct lifestyles, and mechanisms for dealing with toxins. Zebrafish embryos and larvae are commonly used in toxicological studies due to their rapid development. However, this might not fully represent the long-term effects observed in humans over the course of years. As a result, translating findings from zebrafish studies to human risk assessment requires careful consideration of these differences.
- The article should include more recent studies to ensure that the review is current with the latest research in this field. The majority of the references in the current version are from older studies.
Thank you for the valuable comment. The work includes articles published in English from 2012 to 2022. Additionally cited the Chen X, Mou L, Qu J, Wu L, Liu C. Adverse effects of triclosan exposure on health and potential molecular mechanisms. Sci Total Environ. 2023

Round 2
Reviewer 1 Report
review report attached
no major editing is required
Reviewer 2 Report
The authors have addressed all the concerns satisfactorily.